# Defining an intermediate category of tuberculin skin test: A mixture model analysis of two high-risk populations from Kampala, Uganda

Henok G. Woldu[1¤a], Sarah Zalwango[2], Leonardo Martinez[3¤b], María Eugenia Castellanos[1,4]*, Robert Kakaire[1,4], Juliet N. Sekandi[1,4], Noah Kiwanuka[2], Christopher C. Whalen[1,4]

1 Department of Epidemiology and Biostatistics, College of Public Health, University of Georgia, Athens, Georgia, United States of America, 2 Department of Epidemiology and Biostatistics, School of Public Health, College of Health Sciences, Makerere University, Kampala, Uganda, 3 Division of Infectious Diseases and Geographic Medicine, Stanford University School of Medicine, Stanford, California, United States of America, 4 Global Health Institute, College of Public Health, University of Georgia, Athens, Georgia, United States of America

¤a Current address: The Center for Health Analytics for National and Global Equity (C.H.A.N.G.E.), Columbia, Missouri, United States of America
¤b Current address: Department of Epidemiology, School of Public Health, Boston University, Boston, Massachusetts, United States of America
* mecastellanos@uga.edu

**Data Availability Statement:** Data cannot be made publicly available due to ethical restrictions. The

## Abstract

One principle of tuberculosis control is to prevent the development of tuberculosis disease by treating individuals with latent tuberculosis infection. The diagnosis of latent infection using the tuberculin skin test is not straightforward because of concerns about immunologic cross reactivity with the Bacille Calmette-Guerin (BCG) vaccine and environmental mycobacteria. To parse the effects of BCG vaccine and environmental mycobacteria on the tuberculin skin test, we estimated the frequency distribution of skin test results in two divisions of Kampala, Uganda, ten years apart. We then used mixture models to estimate parameters for underlying distributions and defined clinically meaningful criteria for latent infection, including an indeterminate category. Using percentiles of two underlying normal distributions, we defined two skin test readings to demarcate three ranges. Values of 10 mm or greater contained 90% of individuals with latent infection; values less than 7.2 mm contained 80% of individuals without infection. Contacts with values between 7.2 and 10 mm fell into an indeterminate zone where it was not possible to assign infection. We conclude that systematic tuberculin skin test surveys within populations at risk, combined with mixture model analysis, may be a reproducible, evidence-based approach to define meaningful criteria for latent tuberculosis infection.

IRB approval for this study restricts the sharing of individual-level data. An anonymized dataset is available upon request form researchers who meet the criteria for access to confidential information. Data requests may be sent to the Human Subjects Office Director at University of Georgia, Kim Fowler (phone contact: 706-542-5318, and email contact: irb@uga.edu). In particular, we welcome researchers willing to create a strong data-sharing partnership and collaboration with the Ugandan researchers who generated the data.

**Funding:** CCW: R01 AI093856, NO1-AI95383, D43- TW01004. National Institute of Allergy and Infectious Diseases of the National Institutes of Health. https://www.niaid.nih.gov/ The funders had no role in study design, data collection and analysis, decision to publish, or preparation of the manuscript. MEC: Schlumberger Foundation Faculty for the Future Fellowship. No Grant Number. https://www.slb.com/who-we-are/schlumberger-foundation The funders had no role in study design, data collection and analysis, decision to publish, or preparation of the manuscript.

**Competing interests:** The authors have declared that no competing interests exist.

## Introduction

According to the World Health Organization (WHO), the global burden of tuberculosis peaked in 2000 and has since declined by 1.5% per year [1]. Although encouraging, this modest progress falls short of the Millennial Development Goals for tuberculosis elimination. In response to this persistent challenge of tuberculosis, the WHO launched its End TB Strategy in 2015 [2] that promotes integrated, patient-centered care and prevention, bold policies and supportive systems, and intensified research and innovation. In September 2018, the United Nations General Assembly held a high-level meeting to build political commitment and multi-sectoral action to eliminate tuberculosis [3].

With this new commitment to tuberculosis control, the Stop TB Partnership and the WHO now advocate for treatment of latent tuberculosis infection as a way to reduce the risk of tuberculosis among individuals at highest risk for disease [4]. Treatment of latent infection confers benefit not only to the individual but may also confer benefit to a population by shrinking the pool of infected individuals at risk for disease progression.

The diagnosis of latent tuberculosis infection is not straightforward because of persistent questions about the accuracy and reliability of the available diagnostic tests. The diagnosis of infection is made by demonstrating an immune response to antigens of *Mycobacterium tuberculosis* in the absence of clinically active tuberculosis disease. The tuberculin skin test (TST) is a century-old method for assessing tuberculosis infection, but it is limited by immunologic cross-reactivity with Bacille Calmette Guerin (BCG) vaccine and environmental mycobacteria [5, 6], by immunologic boosting [7–9] with repeated tests, and effects of immunosuppression [10]. There are also logistical issues in obtaining good quality tuberculin, maintaining the cold chain, and the need for two separate visits from the health worker [11]. The recent development of interferon-gamma release assays (IGRA) has mitigated some of these concerns about TST [12], but the performance of IGRAs in endemic areas is inconsistent and not fully validated [13–16]. Indeed, in settings where immunosuppression may be common, both tests used in tandem may yield the highest sensitivity [17]. As countries scale up their capacity to prevent tuberculosis, it is likely that the TST will remain the mainstay of diagnosis of latent tuberculosis infection because the test is less expensive and more widely available than IGRA.

The use of TST as the method of diagnosis of latent infection is controversial because there is longstanding and ongoing debate about how best to interpret the results of TST in populations where BCG vaccination is widely used. Some have argued that the effect of cross-reactivity due to BCG vaccination may be minimal and does not change the interpretation of the skin test [18], whereas others have argued the opposite [19, 20]. In an effort to parse the effects of BCG and non-tuberculous mycobacteria on the TST result, researchers have used mixture model analysis to separate underlying component distributions attributable to *M. tuberculosis* infections from other non-specific causes [21–26]. A finite mixture model arises when samples are drawn from a population that is a mixture of K (K >1) component populations and is used for estimating heterogeneity in effects [27–29]. Mixture model analysis is an alternative way to estimate the prevalence of latent tuberculosis infection, which can be compared with the criterion-based methods for assigning latent infection. The criterion based standard tuberculin skin test comprises an intracutaneous purified protein derivatives (PPD) 0.01 ml injection into the forearm where the reaction is read 48 to 72 hours later. Based on the individual person's risk exposure, the threshold used to determine the LTB status can be 5 mm, 10 mm or 15 mm [30].

Although mixture model analysis is useful to understand the epidemiology of latent infection, it does not inform the treatment of latent infection in the individual patient. For these clinical decisions, meaningful criteria for latent infection are needed [27]. The purpose of this

study is to use mixture models to estimate the underlying distributions of the TST in urban Kampala, Uganda, and define meaningful diagnostic criteria for latent tuberculosis infection.

## Materials and methods

### Study populations

Two study populations were used for this analysis: the Kawempe Community Health Study and the Lubaga Community Health study. Lubaga and Kawempe are contiguous divisions in Kampala City (Fig 1). Both studies were performed in Kampala, Uganda, by the same investigators and using similar methodologies and similar standard data collection tools (S1 File).

**Kawempe Community Health Study.** This study was described previously [31, 32]. Briefly, tuberculosis index cases were recruited from Old Mulago Hospital from 1995 to 2005 and were determined to be the initial case presenting in the household. All index cases were microbiologically confirmed using sputum microscopy and culture. Index cases were asked to list their household contacts; these household contacts were defined as any individual spending at least seven consecutive days in the same household as the index case in the three months preceding diagnosis. In this study, 1917 household contacts completed a baseline sociodemographic and tuberculosis risk questionnaire and physical examination collecting data on age, sex, relationship to the index, education level, past tuberculosis, and environmental characteristics.

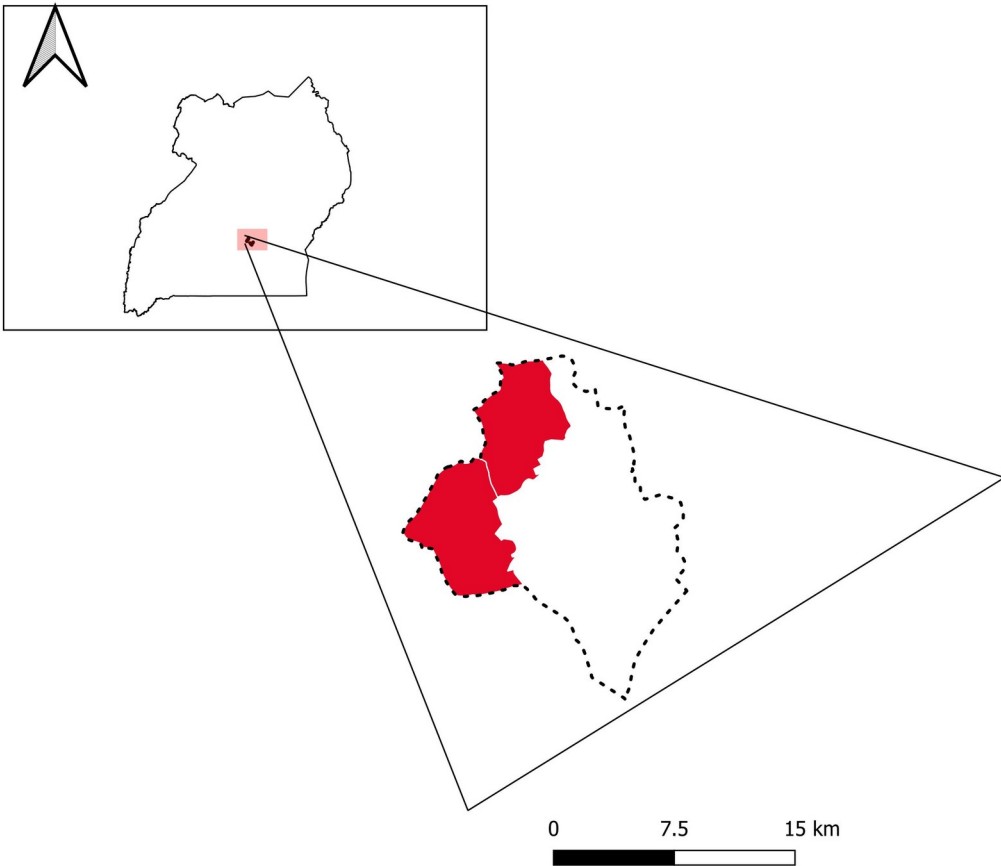

**Fig 1. Geographical locations of study sites: Lubaga and Kawempe divisions (red) within Uganda.** Inset indicates position of study sites within Uganda. Data courtesy of GADM (gadm.org) and Google Earth.

**Lubaga Community Health Study.** This study recruited index tuberculosis cases and disease-free index controls in the Lubaga Division of Kampala, Uganda, between 2012 and 2016. Index tuberculosis cases were recruited from Lubaga Hospital and community health clinics of the Kampala Capital City Authority. All index cases were microbiologically confirmed using sputum smear microscopy, GeneXpert®, or mycobacterial culture. Index controls were matched to the index cases by age category, sex, and neighborhood and were recruited within one month of the matched index case. Index cases and index controls were asked to list household contacts, using the same definition as for the Kawempe study, and contacts who lived outside the household as well. To reduce recall bias we used a combination of standard prompts and recent timeframes to help participants remember their contacts [33]. In this study, there were 1844 contacts of the index cases and controls, 882 and 963 contacts, respectively, who completed a tuberculosis risk questionnaire.

*Measurements*. Many procedures were harmonized between the two studies. All contacts of cases or controls were evaluated for tuberculosis infection using the tuberculin skin test. A TST was performed by placing 0.1 mL of 5 tuberculin units of purified protein derivative on the volar surface of the left forearm of each participant, using the Mantoux method [34]. The induration was independently read by two trained field workers within 48–72 hours using digital calipers to reduce the potential for digit preference bias. If the two indurations were discrepant, an average of the two indurations was used. BCG vaccination was assessed through inspecting deltoid scars and confirmed with medical records when available. Index cases and contacts were tested for HIV infection according to the Ministry of Health Guidelines for Prevention and Treatment of HIV in Uganda (either enzyme-linked immunosorbent assay (Cambridge BioScience, Worcester, MA), or serial rapid tests (Determine HIV-1/2, HIV ½ STAT-PAK, Uni-Gold HIV) [35].

## Ethical approvals

Institutional review board approval was obtained from Ethics Committee at Makerere University School of Public Health and the University of Georgia. Informed consent was obtained from all index cases and controls as well as their contacts. Parents of child contacts provided written consent in addition to child verbal assent.

## Statistical analysis

Frequency distribution and percentages were used to study the baseline characteristics of the study population. Only subjects whose TST induration is $> 0$ mm were included in the mixture models. Visualization of TST induration histogram distribution and the Hartigans' dip test of unimodality [29] were further used to assess whether the TST induration distribution was unimodal or multimodal. Finite mixture of normal models was used to capture the heterogeneity in the TST induration arising from *Mycobacterium tuberculosis* infection or as a result of cross reactions with environmental mycobacteria or prior BCG vaccination [24, 25]. A finite mixture model arises when samples are drawn from a population that is a mixture of K component populations (where K > 1). Let $\lambda i$ represent the proportion of the total population that the *ith* component population constitutes and let $f_i(x)$ represent the probability density function for the *ith* component population. If we represent the measure of the induration size as *X*, a random variable which takes values in the sample space of *w*, its probability density function can be represented as: $g(x) = \lambda 1 f 1(x) \ldots + \lambda k f k(x)$, $x \in w$, $0 \leq \lambda? \leq 1$; $\lambda 1 \ldots + \lambda k = 1$, where $i = 1 \ldots k$ and we say $g(x)$ is a finite mixture of k components. The parameters $\lambda 1 \ldots \lambda k$ are called missing proportions which represent the proportion of the population in each component and

$f1(x)...fk(x)$ are the probability density functions of the random variable X in each component [28].

A method using a combination of Newton-type and expectation-maximization (EM) algorithms was used to estimate parameters of the finite normal mixture models considered. This method was implemented using an R package called "mixdist" [36] in the R programming language (R Core Team). Further, to determine the number of components to be included in our final models, likelihood ratio test and Bayesian Information Criteria (BIC) were used. In both of our study populations, two-component normal mixture models were found to fit the data better than a three-component mixture of normal model or a two component gamma mixture models.

Previous studies have shown that when the class separation in a fitted mixture model is high, a sample size as small as 150 to 300 subjects can perform well [37, 38]. Class separation is at its highest when the difference in the mean between the latent class is large [39]. This study sample size was 3,761 of which the 2051 samples were used in fitting the mixture model.

To assess the effects of age, sex, HIV status and BCG vaccination status on the underlying distributions, we stratified by these variables. We determined the optimal cutoff value for the TST as that TST reading where the two distributions intersect, thereby minimizing misclassification. We stipulated an indeterminate range for the TST induration result by using the 97.5th percentile value of lower distribution and the 2.5th percentile value of higher distribution. The proportion of participants from each group falling in this zone was then calculated. Further sensitivity analysis was carried out to evaluate if those with missing/unknown BCG status differ from those who have BCG status in terms of age, sex, and TST values. However, we did not find any statistical difference between them.

## Results

There were 1,917 participants from Kawempe neighborhood and 1,844 from the Lubaga neighborhood. Although the two groups came from adjacent divisions of Kampala, they differed in several ways (Table 1). The proportion of females was 56% in the Kawempe group and 47% in the Lubaga group. The Lubaga group included a greater proportion of participants in older age groups than Kawempe. In both groups, the coverage of BCG vaccinate was high, 90% in Lubaga and 73% in Kawempe. The proportion of HIV seropositive participants was more than three times higher in the Kawempe than in the Lubaga study population (12% vs 4%).

The mean and median TST induration readings were 11.3 mm (SD = 7.3) and 13.0 mm for Kawempe, and 7.6 mm (SD = 7.5) and 7.4 mm for Lubaga. The difference in the mean and median TST induration between the study populations is attributable to a higher proportion of individuals with a value of 0 mm from the Lubaga group (N = 782, 42%) than from the Kawempe group (N = 381, 20%). Among participants with TST reading > 0 mm, the mean TST induration was 14.1 mm (SD = 5.2) for Kawempe and 13.2 mm (SD = 4.8) for Lubaga. For the Kawempe and Lubaga groups, the frequency distributions of TST among participants with TST induration > 0 mm were multi-modal, according to the Hartigans' dip test of unimodality (D = 0.035, p-value < 0.001; D = 0.019, p-value = 0.01, respectively). A similar multi-modal distribution was found when both cohorts were combined into a single cohort (D = 0.027, p- value < 0.001).

In the Kawempe study, the empirical probability density function of TST distribution among household contacts of index cases was decomposed into two normal distributions using an unstratified mixture model (Table 2, Fig 2A). Using both the likelihood ratio test and BIC criteria, we determined that a model with only two component distributions provided the best fit to the data (Chi square = 71.46, df = 2, p-value < 0.0001). The mean of TST induration

**Table 1. Baseline characteristics of participants in the Kawempe and Lubaga study populations.**

| Category | Kawempe Study | Lubaga Study |
|---|---|---|
| N | 1917 | 1844 |
| Study Periods | 2002–2008 | 2012–2016 |
|  | - - - - - N (%)- - - - - | |
| Sex | | |
| Female | 1064 (55.5) | 867 (47.0) |
| Male | 853 (44.5) | 977 (53.0) |
| Age | | |
| Less than 6 | 524 (27.3) | 177 (9.7) |
| 6 to <16 | 647 (33.8) | 223 (12.2) |
| 16 to < 26 | 597 (31.1) | 627 (34.3) |
| 26 to < 36 | 69 (3.6) | 520 (28.4) |
| ≥ 36 | 80 (4.2) | 281 (15.4) |
| Education* | | |
| None or Primary | 653 (34.2) | 370 (20.1) |
| P2- P8 | 856 (44.9) | 618 (33.5) |
| J1- J2 | 6.0 (0.3) | 0.0 (0.0) |
| S1 to S6 | 355 (18.6) | 707 (38.3) |
| Degree or higher | 37 (1.9) | 149 (8.1) |
| Marital Status* | | |
| Never Married | 1442 (75.9) | 984 (53.8) |
| Married | 324 (17.0) | 601 (32.9) |
| Married Polygamous | 36 (1.9) | 72 (3.9) |
| Divorced | 66 (3.5) | 145 (7.9) |
| Widowed | 33 (1.7) | 26 (1.4) |
| BCG Vaccine* | | |
| No | 498 (27.0) | 153 (9.8) |
| Yes | 1349 (73.0) | 1406 (90.2) |
| HIV Status* | | |
| Positive | 201 (12.2) | 68 (3.9) |
| Negative | 1454 (87.8) | 1695 (96.1) |

Categories with (*) sign have missing observations. From the Kawempe study population: Education = 9 missing, Marital Status = 16 missing, BCG Vaccine = 70 missing and HIV Status = 70 missing. From the Lubaga study: age, education, marital status = 16 missing; BCG vaccine = 200 missing; 85 with unknown HIV serostatus (81 missing).

in the lower distribution was 4.6 mm (SD = 1.9), and the mean for the upper distribution was 15.1 mm (SD = 4.0). The lower distribution comprised 13% and the upper distribution comprised 87% of the population. The optimal cutoff value of the TST for separating the lower and upper distributions was estimated to be 7.1 mm.

In the Lubaga study, the empirical probability density function of TST distribution among both household and non-household contacts of index cases was also decomposed into two normal distributions using an unstratified mixture model (Table 2, Fig 2B). The mean of TST induration in the lower distribution was 7.3 mm (SD = 1.9), and the mean for the upper distribution was 14.7 mm (SD = 4.1). The lower distribution comprised 10% and the upper distribution comprised 90% of the population. The optimal cutoff value of the TST for separating the lower and upper distributions was estimated to be 7.7 mm.

Because the Kawempe and Lubaga studies were performed in the same city, using a similar design and procedures, and because the findings between the two studies were consistent, we

**Table 2. Estimated statistics from fitting a mixture of two normal distributions of TST induration for the Kawempe and Lubaga studies individually and combined into a single, merged study population, then stratified by sex, age group, BCG vaccine status, and HIV infection.**

| | n | $\Pi_1$ | $\mu_1$ | $\sigma_1$ | $\Pi_2$ | $\mu_2$ | $\sigma_2$ | $\mu$ |
|---|---|---|---|---|---|---|---|---|
| Kawempe | 1536 | 0.13 | 4.64 | 1.96 | 0.87 | 15.14 | 4.06 | 7.12 |
| Lubaga | 515 | 0.1 | 7.26 | 1.88 | 0.9 | 14.71 | 4.1 | 7.73 |
| Kawempe and Lubaga combined | 2051 | 0.13 | 5.35 | 2.31 | 0.87 | 15.1 | 4.03 | 7.52 |
| Sex | | | | | | | | |
| Males | 933 | 0.18 | 6.1 | 2.69 | 0.82 | 15.06 | 3.6 | 8.54 |
| Females | 1117 | 0.12 | 5.24 | 2.24 | 0.88 | 15.34 | 4.22 | 7.35 |
| Age Group | | | | | | | | |
| 0–5 | 418 | 0.13 | 3.77 | 1.26 | 0.87 | 14.08 | 3.93 | 5.88 |
| > 5–15 | 611 | 0.16 | 5.12 | 2.55 | 0.84 | 15.19 | 3.63 | 8.04 |
| > 15 | 1022 | 0.15 | 6.99 | 2.49 | 0.85 | 15.68 | 4.11 | 8.64 |
| BCG Vaccine | | | | | | | | |
| Yes | 1522 | 0.18 | 6.19 | 2.7 | 0.82 | 15.27 | 3.76 | 8.62 |
| No | 444 | 0.09 | 4.29 | 1.68 | 0.91 | 15.45 | 4.26 | 6.45 |
| HIV Status | | | | | | | | |
| Positive | 173 | 0.12 | 3.98 | 0.98 | 0.88 | 15.63 | 5.03 | 5.71 |
| Negative | 1631 | 0.17 | 6.32 | 2.76 | 0.83 | 15.36 | 3.76 | 8.60 |

n is for the total number of subjects in each category.

$\Pi_1$ and $\Pi_2$ are the proportion of subjects falling in each component of the mixture of normal model with their respective means and variances given by $\mu_1$, $\mu_2$, $\sigma_1$, $\sigma_2$.

The overall optimal cutoff point for all the data set and each stratified group is shown in the column given by $\mu$.

combined the contacts of index cases from the two populations to form a single study population (Table 2, Fig 3). As seen in the individual studies, the optimal fit was achieved with two normal distributions in the combined analysis. The mean of TST induration in the lower distribution was 5.4 mm (SD = 2.3), and the mean for the upper distribution was 15.1 mm (SD = 4.0). The lower distribution comprised 13% and the upper distribution comprised 87% of the population. The optimal cutoff value for separating the lower and upper distributions was estimated to be 7.5 mm.

When stratifying the population by age category, the mean for the upper distribution was remarkably consistent, ranging from 14.0 mm to 15.7 mm across three different subgroups (Table 2), whereas the mean value for the lower distribution was more variable across age categories, increasing from 3.8 mm to 7.0 mm. A similar pattern was observed when stratifying by HIV serostatus. When stratifying by BCG vaccination, the mean value of the lower distribution was greater among those who were vaccinated compared to those who were not (6.2 mm versus 4.3 mm), but the means of the upper distributions were similar. Age younger than 5 years and HIV infection both had lower optimal cutoff values compared to their counterparts, whereas BCG vaccination required a higher cutoff value.

In the Lubaga study, but not the Kawempe study, we included contacts of index controls who did not have tuberculosis. Among 556 household contacts of index controls who were tested with TST, the distribution of the control contacts decomposed into two normal distributions with means of 6.8 mm (s.d. = 2.3) and 14.3 mm (s,d, = 4.1). Twenty-five percent of contacts fell under the lower curve, whereas 75% fell under the upper curve.

From the combined analysis, we evaluated the overlap of the two distributions (Table 3). The value of 9.9 mm represented the 97.5th percentile value for the lower distribution, so only 2.5% of this distribution exceeded this value, whereas 90.2% of the upper distribution fell

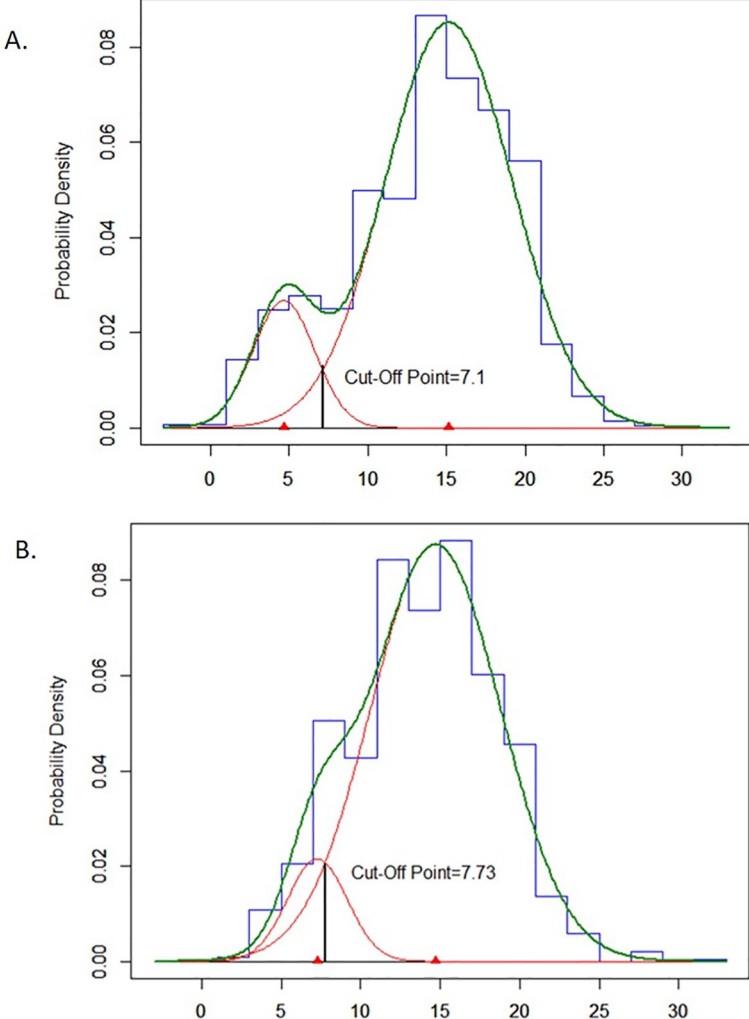

**Fig 2. Estimated underlying normal distributions of TST induration among household contacts on index cases for the Kawempe study (A, N = 1536) and among household and non-household contacts of index cases for the Lubaga study (B, N = 515).**

above this value. Similarly, the value of 7.2 mm represented the 2.5th percentile of the upper distribution, so only 2.5% fell below this value, whereas 78.9% of the lower distribution fell below this cutoff value. The interval of 7.3 mm to 9.8 mm contained 18.6% of the lower distribution and 7.3% of the upper distribution. A TST value falling in this range was 2.5 times more likely to fall under the lower distribution.

## Discussion

In this analysis, we compared the TST responses from two independent groups of tuberculosis contacts in Kampala, Uganda, from two time periods about one decade apart. We found that the overall frequency distribution for each group could be decomposed into two normal distributions that had remarkably similar mean TST values between the two groups. When the two groups were combined, we found that the mean TST value for the upper distribution was 15 mm and stable across strata of sex, age group, BCG vaccination, and HIV serostatus, whereas

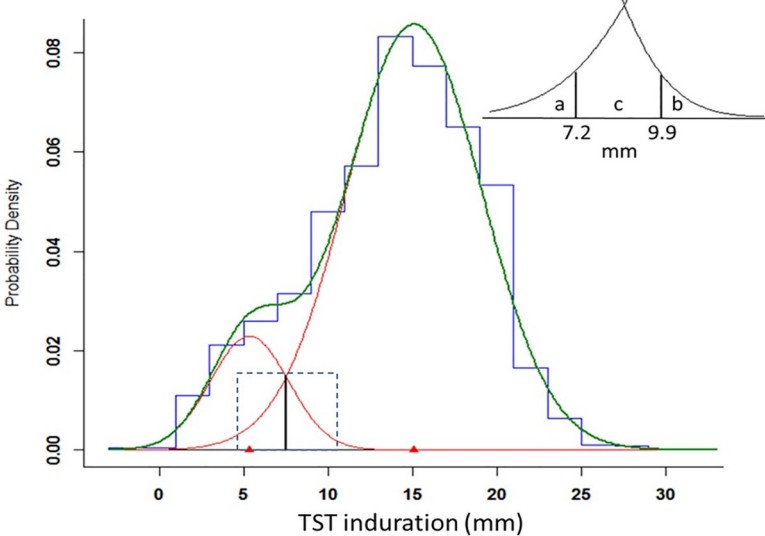

**Fig 3. Estimated normal mixture distributions of TST induration among contacts of index cases from a single, merged study population of the Kawempe and Lubaga studies (N = 2051).** Vertical solid line represents optimal cutoff of 7.5 mm between two distributions. Dashed box delineates area of INSERT: a contains 2.5% of the upper distribution; b contains 2.5% of the lower distribution; c contains 18.6% of the lower distribution and 7.3% of the upper distribution.

the mean TST value for the lower distribution varied by age group, HIV serostatus and BCG vaccination.

We posit that the upper distribution represents true infection with *M. tuberculosis* whereas the lower represents the non-specific effects from BCG vaccination, nontuberculous mycobacteria infection, or immune modulation. Since there is no 'gold standard' for latent tuberculosis infection, we rely on other ways to establish the validity of this assertion. For the upper distribution, there is content validity since the mean TST value of 15 mm for the upper distribution is entirely consistent with the skin test results of patients with active tuberculosis disease [40]. It is also consistent with similar studies of tuberculosis infection from South Korea, Malawi, and the Basque region of Spain [21, 24, 26]. Moreover, the cutoff value of 9.9 mm is essentially the same as the 10 mm criterion recommended to assign latent tuberculosis infection [41]. Like the TST survey from Malawi [21], we found the frequency distribution of reactive tests was indeed best described by two underlying distributions with mean values nearly 10 mm different (5.4 versus 15.1 mm). Perhaps most important, a greater proportion of contacts from the homes of tuberculosis cases fell under the upper distribution as compared to the contacts from control homes (87% versus 75%, respectively).

As for the lower distribution, we propose that it represents sensitization from infection with environmental mycobacteria or prior BCG vaccination. To assess the effect of environmental mycobacteria, we evaluated contacts without BCG vaccination and found that the mean value

**Table 3. Overlap between the lower and upper distributions and interpretation of TST in the entire, combined study population.**

| TST Cutoff Value (mm) | Lower Distribution | Upper Distribution | Interpretation |
|---|---|---|---|
| | - - - - - %- - - - - | | |
| 7.2 | 78.9 | 2.5 | Negative below 7.2 mm |
| 7.3–9.8 | 18.6 | 7.3 | Indeterminate |
| 9.9 | 2.5 | 90.2 | Positive above 9.9 mm |

of the lower distribution was 4.3 mm, which is consistent with an effect from environmental mycobacteria infection in the population. There also appeared to be a small effect of BCG vaccination because the mean value of the lower distribution was 1.9 mm larger among vaccinated contacts compared with those who were not vaccinated. We acknowledge that it is difficult to parse these effects without firm epidemiologic information about environmental mycobacterial infection or specialized immunologic tests to evaluate immunity to BCG. It is also possible that by not including community controls from the Kawempe study, we have underestimated the number of TST responses with low values from that study.

Mixture models have been used to estimate the prevalence of latent tuberculosis infection without using a defined criterion for infection. Typically, authors have used the upper distribution as an indicator of infection and used the proportion of the population under this curve as the prevalence. Although this type of mixture modeling is useful in understanding the epidemiology of tuberculosis infection in a population, the findings from these models are not readily applied to the clinical setting because they do not infer criteria to define latent infection.

To guide clinical decisions regarding treatment of latent tuberculosis infection, it is customary to interpret the TST as a dichotomous test—either positive or negative. Since the TST is inherently context-dependent [21], the criteria for a positive test may vary depending on age distribution, environmental mycobacteria, co-morbidities such as HIV infection, and recent exposure. Indeed, our findings reflect this variability and support the use of different cutoff values for different populations, as is currently practiced. For example, if we use the optimal cutoff values for separating the lower and upper distributions as our criterion for infection, then we would propose 6 mm as the criterion for infection in children younger than 5 years and in HIV infection. The criterion for infection among contacts with BCG vaccination is nearly 9 mm, which is within the margin of error of a 10 mm reaction.

Creating a dichotomous criterion for the TST ignores potentially useful information found in the continuous measurement [25] and may lead to misclassification of latent infection. We propose another interpretation of the TST, one that categorizes the TST results, but adds a third indeterminate category to account for some of the uncertainty in the TST. Using percentiles of the underlying normal distributions estimated by the mixture model, we defined two TST values to demarcate three ranges. Values of 9.9 mm or greater contained 90% of individuals with latent tuberculosis infection (the upper distribution), whereas TST values less than 7.2 mm contained nearly 80% of individuals without tuberculosis infection (lower distribution). Contacts with values between 7.2 and 9.8 mm fell into an indeterminate zone where it was not possible to classify them as infected or not. In our sample, contacts with responses in this zone were 2.5 times more likely to fall under the lower distribution than the upper distribution, so were more likely to represent reactions resulting from BCG vaccination or infection with other mycobacteria.

By defining this third category, we acknowledge that TST readings in this range are uncertain, but we preserve clinically useful cutoff values and gain clarity about the interpretation of readings outside of the indeterminate range. As for what to do with a person who has an indeterminate reading, it may depend on the age, HIV serostatus, or history of recent exposure. But for the many adult individuals who have no known exposure, we propose repeating the test in 2 to 4 weeks. With the repeat test, we expect regression toward the mean, so subsequent readings would migrate toward the true underlying distribution or a booster response [8]. These movements might help guide decisions for treatment. An alternative approach would be to perform sequential testing, first with the TST followed by the interferon- γ release assay for tests within the indeterminate range, and base decisions about latent infection on the results of these tests together [25].

With the advent of IGRAs, one may argue that the TST is obsolete and refined criteria for infection using the TST are no longer needed. Although the scientific justification of IGRAs is strong [12], the performance characteristics of these tests are not optimal or consistent in some populations, especially in Africa and Asia [14, 42, 43]. The c-TB (Statens Serum Institute, Copenhagen, Denmark) is a new skin test based on ESAT-6 and CFP-10 antigens, the same antigens used in the current IGRAs, that appears to be unaffected by BCG vaccination [40]. If the c-TB skin test performs well in African populations where tuberculosis is endemic and BCG vaccination is widely used, it may replace skin tests using purified protein derivative. Until then, the proposed modification to the TST in assigning latent infection may be useful in decisions to treat latent infection or revaluate after continued follow-up.

We do not presume to suggest that there are fixed or standard criteria that define latent tuberculosis infection across populations. As has been pointed out by others, population characteristics and the goals of testing affect the choice of cutoff values [21, 44–46]. We do propose, however, that the process of TST surveys within populations at risk, followed by a mixture model analysis, is an evidence-based approach that can define meaningful criteria for latent infection in a given population.

## Supporting information

**S1 File. Data collection of participant's characteristics (English and Luganda versions).** (DOCX)

## Acknowledgments

The authors would like to thank the Uganda National Tuberculosis and Leprosy Control Programme, the municipal leaders of Lubaga Division of Kampala, and the Kampala Capital City Authority for their cooperation and support of the field work needed to complete this study. The authors acknowledge the home health visitors for invaluable contributions to the science of this study and the Epidemiology in Action research group of the Global Health Institute, University of Georgia, for their critical insights.

## Author Contributions

**Conceptualization:** Henok G. Woldu, Robert Kakaire, Noah Kiwanuka, Christopher C. Whalen.

**Data curation:** Sarah Zalwango, Robert Kakaire, Noah Kiwanuka, Christopher C. Whalen.

**Formal analysis:** Henok G. Woldu, Leonardo Martinez, María Eugenia Castellanos, Robert Kakaire, Juliet N. Sekandi, Noah Kiwanuka, Christopher C. Whalen.

**Funding acquisition:** Noah Kiwanuka, Christopher C. Whalen.

**Methodology:** Henok G. Woldu.

**Project administration:** Robert Kakaire.

**Supervision:** Christopher C. Whalen.

**Writing – original draft:** Henok G. Woldu, Christopher C. Whalen.

**Writing – review & editing:** Henok G. Woldu, Sarah Zalwango, Leonardo Martinez, María Eugenia Castellanos, Robert Kakaire, Juliet N. Sekandi, Noah Kiwanuka, Christopher C. Whalen.

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
