## [Decision Letter · Decision Letter 0]

1 Sep 2020

PONE-D-20-22617

DEFINING AN INDETERMINATE TUBERCULIN SKIN TEST: A MIXTURE MODEL ANALYSIS OF TWO HIGH-RISK POPULATIONS FROM KAMPALA, UGANDA

PLOS ONE

Dear Dr.ssa Castellanos,

Thank you for submitting your manuscript to PLOS ONE. After careful consideration, we feel that it has merit but does not fully meet PLOS ONE’s publication criteria as it currently stands. Therefore, we invite you to submit a revised version of the manuscript that addresses the points raised during the review process.

We look forward to receiving your revised manuscript.

Kind regards,

Francesco Di Gennaro

Academic Editor

PLOS ONE

Journal Requirements:

2. Please address the following:

a)  Please include additional information regarding the survey or questionnaire used in the study and ensure that you have provided sufficient details that others could replicate the analyses. For instance, if you developed a questionnaire as part of this study and it is not under a copyright more restrictive than CC-BY, please include a copy, in both the original language and English, as Supporting Information. In addition, please provide further details concerning the development of this tool.

b)  Please ensure you have thoroughly discussed any potential limitations of this study within the Discussion section, including the potential introduction of baises during data collection.

c)  Please describe the reasons for any missing data, if known.

Additional Editor Comments (if provided):

Dear Authors, follow reviewer indication to improve your article

Reviewers' comments:

Reviewer's Responses to Questions

**Comments to the Author**

1. Is the manuscript technically sound, and do the data support the conclusions?

Reviewer #1: Yes

Reviewer #2: Partly

2. Has the statistical analysis been performed appropriately and rigorously? 

Reviewer #1: Yes

Reviewer #2: Yes

3. Have the authors made all data underlying the findings in their manuscript fully available?

Reviewer #1: Yes

Reviewer #2: No

4. Is the manuscript presented in an intelligible fashion and written in standard English?

Reviewer #1: Yes

Reviewer #2: Yes

5. Review Comments to the Author

Reviewer #1: In this study entiled: Defining an indeterminate tuberculin skin test: a mixture model analysis of two high-risk populations from Kampala, Uganda

This is a well-intended initiative to address a major problem affecting the African population.

1The study, context is in keeping with other literature on the topic. Latent TB remains a pool for active TB disease and treatment of latent TB would be a better elimination strategy.

2 Indeed the readily available TST is not well interpreted because of cross reaction of the BCG vaccine and non tuberculosis bacteria. Consequently the identification of the true latent infected individuals may sometimes be missed out. The other tests like IGRA for latent TB diagnosis is not validated endemic areas. Therefore the most popular test may remain TST.

3 The authors therefore set out to map out the epidemiological distribution of TST in two study populations of urban Kampala using mixture model method and to define criteria for diagnosis of the TST test.

4 The selected study populations were appropriate, Kawempe division one of the crowding areas of Kampala yet crowding is among the well known risk factors for mtb transmission.

5 Sample size and study duration for both studies were adequate to generate sufficient data for the analysis.

6 The mixture model method used was appropriate since they were dealing with interactable variables and data was from two different populations.

The Lubaga division included controls in their study. The Kawempe study however did not include controls.

7Hartigans’ dip test of unimodality to asses distribution was unimodal or multimodal was appropriate .

8 The Finite mixture normal model was for heterogeneity was appropriate.

9 presentations of data available is appropriate with tables and figures supporting it

10The authors were able to determine an intermediate reading they referred to as indeterminate.

Reviewer’s comments

In view of the above observations, the this study, Defining an indeterminate tuberculin skin test: a mixture model analysis of two high-risk populations from Kampala, Uganda

I am only concerned about the Kawempe study that did not have a control group.

Are these findings generalized to other populations at risk of disease?

On the whole I find this study appropriate for publication

Reviewer #2: This manuscript clearly defines a new category of clinical relevance in the application of the highly pervasive tuberculin skin test for tuberculosis in resource poor settings. This new category introduces greater rigor and robustness in the algorithm for decision making in regard to committing a borderline subject to tuberculosis treatment or not by proposing a follow up step for those in this category.

General Comments

1. Your work is technically sound and received the rigour it deserves, by clearly describing the subject recruitment procedure. You, however, mention completely nothing about the appropriateness of your sample size with reference to the characteristics of the study population. A simple map to illustrate the relative locations of Lubaga and Kawempe within Kampala City would provide some spatial clarity to your audience. I suggest that you consider providing these pieces of information for benefit of the audience.

2. The data underlying the findings have not been made available. Was it not possible to deprecate the identities of the study subjects and achieve anonymity?

Specific Comments

Title

After reading your manuscript a couple of times, I suggest it should read as: "Defining an Intermediate Category of Tuberculin Skin Test: A Mixture Model Analysis of Two High-Risk Populations From Kampala, Uganda"

Abstract

line #27 please note that Kampala does not have districts within, the administrative units of Lubaga and Kawempe are officially referred to as "divisions".

Introduction

line #78 Please consider a brief explanation of what the Mixture Model Analysis involves.

line #79 consider explaining the concept of criterion based methods.

Alternatively, consider clearly acknowledging the authorities of these two concepts in lines #78 and #79.

Materials and Methods

line #123 please find and appropriately acknowledge the Ministry of Health Guidelines on HIV.

line #150 EM appears for the first time as an abbreviation. Consider writing this in full at first mention with the abbreviation in brackets.

line #152 consider revising to read as "......"mixdist"(33) in the R programming language (R Core Team)".

Discussion

line #290 consider deleting ".......the other....". these appear redundant and do not add clarity to your point here.

line #346 what is the rationale for this period? This is certainly not deduced from your data and analysis.

line #364 please provide the authorities referred to here as "......others"

6. PLOS authors have the option to publish the peer review history of their article (what does this mean?). If published, this will include your full peer review and any attached files.

Reviewer #1: **Yes: **ESTER LILIAN ACEN

Reviewer #2: No

---

## [Author Response · Author response to Decision Letter 0]

12 Oct 2020

RESPONSE TO EDITOR AND REVIEWERS (We also uploaded an attached file with this same information)

PONE-D-20-22617

DEFINING AN INDETERMINATE TUBERCULIN SKIN TEST: A MIXTURE MODEL ANALYSIS OF TWO HIGH-RISK POPULATIONS FROM KAMPALA, UGANDA

PLOS ONE

Response: We appreciate the careful review and thoughtful comments of our manuscript. In our revision, we have made every effort to clarify the concerns raised by you and the reviewers and to respond accordingly. Please note that the line numbers we provided correspond to the revised clean version of the manuscript.

and 

R/We have revised those requirements and made changes to meet PLOS ONE’s style requirements.

 2. Please address the following:

a) Please include additional information regarding the survey or questionnaire used in the study and ensure that you have provided sufficient details that others could replicate the analyses. For instance, if you developed a questionnaire as part of this study and it is not under a copyright more restrictive than CC-BY, please include a copy, in both the original language and English, as Supporting Information. In addition, please provide further details concerning the development of this tool.

R/We have included as Supporting File 1 the standard questionnaire we used in the study for the collection of the main participant’s characteristics (English and Luganda version). We have added this information in the manuscript (lines 96-98). 

b) Please ensure you have thoroughly discussed any potential limitations of this study within the Discussion section, including the potential introduction of biases during data collection.

R/ We now have included measures we took to reduce recall bias during the data collection (lines 122-124). Moreover, from line 329 to line 331, we have discussed the potential limitations of our study, including biases regarding our study population. 

c) Please describe the reasons for any missing data, if known.

R/ All subjects included in our analysis had the tuberculin skin test (TST) result which was main outcome of interest. For some subjects, there is missing covariate information. The dataset for this analysis, which was finalized in 2017, had missing values of education, marital status, HIV status, and BCG vaccine status. Information collected through self-report may result from the refusal of the study participant to answer the question. HIV testing was not indicated on all study participants, so was obtained using a standard protocol. As for BCG vaccination, we collected this variable in overlapping ways. We examined each person for a BCG scar and verified the presence of scar with vaccination card where available. Most participants younger than 20 years had a vaccination card. For adults, we relied on verbal confirmation that they had received a BCG vaccination when younger. We have updated the footnote of Table 1 to explain that for the 285 subjects with missing BCG result; 85 of these 285 answered that they did not know whether they had received the vaccine or not. To evaluate the effect of this missing information, we performed a sensitivity analysis and found that those with and without BCG information did not differ as regards to age, sex, and TST values (lines 185-187).

Data Availability: Data cannot be made publicly available due to ethical restrictions. The IRB approval for this study restricts the sharing of individual-level data. An anonymized dataset is available upon request form researchers who meet the criteria for access to confidential information. Data requests may be sent to the Human Subjects Office Director at University of Georgia, Kim Fowler (phone contact: 706-542-5318, and email contact: irb@uga.edu). In particular, we welcome researchers willing to create a strong data-sharing partnership and collaboration with the Ugandan researchers who generated the data.

5. Review Comments to the Author

Reviewer #1: In this study entitled: Defining an indeterminate tuberculin skin test: a mixture model analysis of two high-risk populations from Kampala, Uganda

This is a well-intended initiative to address a major problem affecting the African population.

1The study, context is in keeping with other literature on the topic. Latent TB remains a pool for active TB disease and treatment of latent TB would be a better elimination strategy.

2 Indeed the readily available TST is not well interpreted because of cross reaction of the BCG vaccine and non tuberculosis bacteria. Consequently the identification of the true latent infected individuals may sometimes be missed out. The other tests like IGRA for latent TB diagnosis is not validated endemic areas. Therefore the most popular test may remain TST.

3 The authors therefore set out to map out the epidemiological distribution of TST in two study populations of urban Kampala using mixture model method and to define criteria for diagnosis of the TST test.

4 The selected study populations were appropriate, Kawempe division one of the crowding areas of Kampala yet crowding is among the well known risk factors for mtb transmission.

5 Sample size and study duration for both studies were adequate to generate sufficient data for the analysis.

6 The mixture model method used was appropriate since they were dealing with interactable variables and data was from two different populations.

The Lubaga division included controls in their study. The Kawempe study however did not include controls.

7Hartigans’ dip test of unimodality to asses distribution was unimodal or multimodal was appropriate .

8 The Finite mixture normal model was for heterogeneity was appropriate.

9 presentations of data available is appropriate with tables and figures supporting it

10The authors were able to determine an intermediate reading they referred to as indeterminate.

Reviewer’s comments

In view of the above observations, this study, Defining an indeterminate tuberculin skin test: a mixture model analysis of two high-risk populations from Kampala, Uganda

R/We thank and appreciate the favorable review to our manuscript.

I am only concerned about the Kawempe study that did not have a control group.

R/We agreed and have included as limitation of the study that we do not have control group for the Kawempe study (lines 329-331).

Are these findings generalized to other populations at risk of disease?

R/We included the following paragraph as our closing paragraph (lines 387-392): “We do not presume to suggest that there are fixed or standard criteria that define latent tuberculosis infection across populations. As has been pointed out by others, population characteristics and the goals of testing affect the choice of cutoff values. We do propose, however, that the process of TST surveys within populations at risk, followed by a mixture model analysis, is an evidence-based approach that can define meaningful criteria for latent infection in a given population.”

On the whole I find this study appropriate for publication

Reviewer #2: This manuscript clearly defines a new category of clinical relevance in the application of the highly pervasive tuberculin skin test for tuberculosis in resource poor settings. This new category introduces greater rigor and robustness in the algorithm for decision making in regard to committing a borderline subject to tuberculosis treatment or not by proposing a follow up step for those in this category.

General Comments

1. Your work is technically sound and received the rigour it deserves, by clearly describing the subject recruitment procedure. You, however, mention completely nothing about the appropriateness of your sample size with reference to the characteristics of the study population. A simple map to illustrate the relative locations of Lubaga and Kawempe within Kampala City would provide some spatial clarity to your audience. I suggest that you consider providing these pieces of information for benefit of the audience.

R/Thank you very much for your comments. We have included a paragraph describing the appropriateness of our sample size for this analysis (lines 173-177). Also, we have included a map (Figure 1) illustrating the relative locations of Lubaga and Kawempe Divisions.

2. The data underlying the findings have not been made available. Was it not possible to deprecate the identities of the study subjects and achieve anonymity?

R/ Data cannot be made publicly available due to ethical restrictions. The IRB approval for this study restricts the sharing of individual-level data. An anonymized dataset is available upon request form researchers who meet the criteria for access to confidential information. Data requests may be sent to the Human Subjects Office Director at University of Georgia, Kim Fowler (phone contact: 706-542-5318, and email contact: irb@uga.edu). In particular, we welcome researchers willing to create a strong data-sharing partnership and collaboration with the Ugandan researchers who generated the data.

Specific Comments

Title

After reading your manuscript a couple of times, I suggest it should read as: "Defining an Intermediate Category of Tuberculin Skin Test: A Mixture Model Analysis of Two High-Risk Populations From Kampala, Uganda"

R/We appreciate the suggestion and we have modified our title (line 1 and line 2). 

Abstract

line #27 please note that Kampala does not have districts within, the administrative units of Lubaga and Kawempe are officially referred to as "divisions".

R/We agreed with this suggestion. We have changed the term in line 26.

Introduction

line #78 Please consider a brief explanation of what the Mixture Model Analysis involves.

R/We have included a brief explanation of the mixture model analysis (lines 77-79) in the Introduction Section, added references and we have provided a detailed explanation in the Methods Section (lines 154 -164). 

line #79 consider explaining the concept of criterion based methods.

Alternatively, consider clearly acknowledging the authorities of these two concepts in lines #78 and #79.

R/We have expanded on the criterion for assigning latent infection using tuberculin skin test and we have provided appropriate references (lines 81-85). 

Materials and Methods

line #123 please find and appropriately acknowledge the Ministry of Health Guidelines on HIV.

R/We have renamed and made the proper reference the Guidelines for Prevention and Treatment of HIV in Uganda (lines 137-138, 140). 

line #150 EM appears for the first time as an abbreviation. Consider writing this in full at first mention with the abbreviation in brackets.

R/We have followed the suggestion and mention the full name of the expectation-maximization ‘EM’ algorithm (lines 165-166).

line #152 consider revising to read as "......"mixdist"(33) in the R programming language (R Core Team)".

R/We have revised and changed as suggested (lines 167-168).

Discussion

line #290 consider deleting ".......the other....". these appear redundant and do not add clarity to your point here.

R/We have revised and deleted this term as suggested (line 311).

line #346 what is the rationale for this period? This is certainly not deduced from your data and analysis.

R/We have expanded the rationale for this period, explaining that after 2-4 weeks, a second test will show a booster response or a regression toward the mean (lines 369-371).

line #364 please provide the authorities referred to here as "......others"

R/We have added the corresponding references (389-390).

---

## [Decision Letter · Decision Letter 1]

29 Dec 2020

Defining an intermediate category of tuberculin skin test: A mixture model analysis of two high-risk populations from Kampala, Uganda

PONE-D-20-22617R1

Dear Dr. Castellanos,

We’re pleased to inform you that your manuscript has been judged scientifically suitable for publication and will be formally accepted for publication once it meets all outstanding technical requirements.

Kind regards,

Francesco Di Gennaro

Academic Editor

PLOS ONE

Additional Editor Comments (optional):

dear authors congratulations

Reviewers' comments:

Reviewer's Responses to Questions

**Comments to the Author**

1. If the authors have adequately addressed your comments raised in a previous round of review and you feel that this manuscript is now acceptable for publication, you may indicate that here to bypass the “Comments to the Author” section, enter your conflict of interest statement in the “Confidential to Editor” section, and submit your "Accept" recommendation.

Reviewer #1: All comments have been addressed

2. Is the manuscript technically sound, and do the data support the conclusions?

Reviewer #1: Yes

3. Has the statistical analysis been performed appropriately and rigorously? 

Reviewer #1: Yes

4. Have the authors made all data underlying the findings in their manuscript fully available?

Reviewer #1: No

5. Is the manuscript presented in an intelligible fashion and written in standard English?

Reviewer #1: Yes

6. Review Comments to the Author

Reviewer #1: My concerns have been addressed accordingly

We agreed and have included as limitation of the study that we do not have control group for the Kawempe study (lines 329-331).

(lines 387-392): “We do not presume to suggest that there are fixed or standard criteria that define latent tuberculosis infection across populations. As has been pointed out by others, population characteristics and the goals of testing affect the choice of cutoff values. We do propose, however, that the process of TST surveys within populations at risk, followed by a mixture model analysis, is an evidence-based approach that can define meaningful criteria for latent infection in a given population.

7. PLOS authors have the option to publish the peer review history of their article (what does this mean?). If published, this will include your full peer review and any attached files.

Reviewer #1: **Yes: **ESTER LILIAN ACEN

---

## [Editor Report · Acceptance letter]

13 Jan 2021

PONE-D-20-22617R1 

Defining an intermediate category of tuberculin skin test: A mixture model analysis of two high-risk populations from Kampala, Uganda 

Dear Dr. Castellanos:

I'm pleased to inform you that your manuscript has been deemed suitable for publication in PLOS ONE. Congratulations! Your manuscript is now with our production department. 

Kind regards, 

on behalf of

Dr. Francesco Di Gennaro 

Academic Editor

PLOS ONE